# Crystallization and Microstructure Evolution of Microinjection Molded Isotactic Polypropylene with the Assistance of Poly(Ethylene Terephthalate)

**DOI:** 10.3390/polym12010219

**Published:** 2020-01-16

**Authors:** Zhongguo Zhao, Xin Zhang, Qi Yang, Taotao Ai, Shikui Jia, Shengtai Zhou

**Affiliations:** 1National and Local Engineering Laboratory for Slag Comprehensive Utilization and Environment Technology, School of Materials Science and Engineering, Shaanxi University of Technology, Hanzhong 723000, China; zhaozhongguo@snut.edu.cn (Z.Z.); zhangxinsimilar@163.com (X.Z.); aitaotao0116@126.com (T.A.); shikuijiagd@163.com (S.J.); 2The State Key Laboratory of Polymer Materials Engineering, College of Polymer Science and Engineering, Sichuan University, Chengdu 610065, China; 3The State Key Laboratory of Polymer Materials Engineering, Polymer Research Institute, Sichuan University, Chengdu 610065, China

**Keywords:** immiscible blends, microstructure, microinjection molding, microfibrils

## Abstract

In this work, a series of isotactic polypropylene/poly(ethylene terephthalate) (iPP/PET) samples were prepared by microinjection molding (μIM) and mini-injection molding (IM). The properties of the samples were investigated in detail by differential scanning calorimetry (DSC), Wide-Angle X-ray Diffraction (WAXD), Polarized light microscope (PLM) and scanning electron microscopy (SEM). Results showed that the difference in thermomechanical history between both processing methods leads to the formation of different microstructures in corresponding iPP/PET moldings. For example, the dispersed spherical PET phase deforms and emerges into continuous in-situ microfibrils due to the intensive shearing flow field and temperature field in μIM. Additionally, the incorporation of PET facilitates both the laminar branching and the reservation of oriented molecular chains, thereby leading to forming a typical hybrid structure (i.e., fan-shaped β-crystals and transcrystalline). Furthermore, more compact and higher degrees of oriented structure can be obtained via increasing the content of PET. Such hybrid structure leads to a remarkable enhancement of mechanical property in terms of μIM samples.

## 1. Introduction

Increasing attention has been paid to the miniaturization of products or components in the areas of electronics, biomedical and microelectromechanical systems, due to the fact that more and more functions are being integrated into smaller spaces [1,2,3,4,5]. Microinjection molding (μIM) has been widely adopted to prepare microcomponents, which features very high shearing and cooling effects during the mold cavity filling process [4,6,7]. Relative to the microparts made with metals, polymeric microparts can be produced at a large scale at relatively low cost per part [8]. 

The mechanical properties of isotactic polypropylene (iPP) are relatively low, and this characteristic cannot satisfy the demands for engineering applications. Thus, incorporating reinforcing fibers into iPP is recurrently adopted to enhance its mechanical properties [9,10,11,12]. One of the most frequently used methods is blending the iPP matrix with glass fibers and poly(ethylene terephthalate) (PET) microfibrils [13,14,15,16]. 

The in-situ PET microfibrils can be formed through a “die-extrusion, hot-stretching, cooling” technique. [17] The in-situ PET microfibrillar morphology can be significantly affected by the hot-stretching ratio and PET concentration. However, the introduction of PET microfibrils or glass fibers into the polymers can significantly increase the melt viscosity, resulting in the decrease of filling property. Developing high performance microparts with enhanced interfacial properties seems critical for μIM. In the present work, μIM is adopted to adjust the microstructure of iPP/PET blends, and so called “in-situ PET microfibrils” can be formed during this μIM process. In our previous report [17,18], the formation of PET microfibers was affected by the mold temperature and melt temperature under microinjection molding conditions. Both increasing the melt temperature and decreasing the mold temperature are beneficial to form well-defined, long, PET microfibers. However, the formation mechanism about the effect of mold temperature and PET concentration upon the in-situ formed PET microfibrils is obscure. 

Relative to conventional injection molding, the inherent thermomechanical history in IM (i.e., very high shearing and cooling effects) leads to forming different phase structure and crystalline structure, which can significantly affect the final properties of subsequent moldings. The advantages of utilizing fibers to strengthen iPP lie in not only their excellent mechanical properties, but also their strong capability of tailoring the crystalline structure. It is known that the oriented crystalline structure occurs during injection molding and/or shearing hot stage treatment. The formed structures, such as the shish–kebab structure or transcrystallinity perpendicular to the axis of fibers will distinctly affect the final properties of samples [19,20,21]. Unlike conventional injection molding, the melt flow in μIM is quite complex, and the classical “foundation flow” model might not hold in such a complicated scenario. Hence, μIM is becoming a platform to study the crystallinity and evolution of phase structure, such as supermolecular structures (shish-kebab, transcrystallite, or β-cylindrite), in a complex flow field. 

The μIM process is quite complex, and many factors such as the high shear stress, mold temperature, cooling-rate etc., can greatly affect the properties of the microparts. The alteration in processing conditions leads to forming different microstructure and crystalline structure, thereby affecting the mechanical properties of the resulting microparts. Generally, mold temperature, shear rate and PET concentration have a significant effect on governing the final microfeatures of an in-situ microfibrillar PET/polyolefin blend. A dramatic enhancement of crystallization kinetics can be also caused by the application of the shear flow field and PET microfibrils. For example, Jin et al. [22] studied the changes of crystalline structures in microinjection-molded isotactic polypropylene/β-nucleating agent samples, and they fund that amounts of the stratiform β-crystals’ structure exists. The above shows that applying extreme shear flow and a temperature field is prone to alter the evolution of the microstructure. Thus, it is necessary and meaningful to investigate the effect of blend ratio on the evolution of hierarchical crystalline structure and phase morphology in iPP/PET immiscible blends. 

In this current study, the mechanism of the effect of mold temperatures and blend ratio upon the morphology evolution and the crystallization behavior was studied, revealing the relationship between the processing conditions and properties of iPP/PET blends. Comparison of the morphology and evolution of microstructure at different mold temperatures and PET concentrations between μIM microparts and IM macroparts was studied using SEM, PLM, DSC, Small-Angle X-ray Scattering (SAXS) and the WAXD technique. 

## 2. Experimental

### 2.1. Materials

The isotactic polypropylene (iPP) used as the matrix was T30S. A commercial product with M_w_ of about 587,000 g/mol was provided by Lanzhou Petroleum Chemical Co. (Lanzhou, China). Poly(ethylene terephthalate) (PET) was kindly provided by LuoYang Petroleum Chemical Co., LuoYang, China. 

### 2.2. Sample Preparation

Firstly, iPP was mixed with various PET content using an internal mixer (XSS-300) at 55 rpm and 270 °C for 5 min, and then, the iPP/PET blends were injected into microparts and macroparts with the melting temperature of 270 °C and mold temperature of 80 and 120 °C by using microinjection molding (Micropower5, Battenfeld GmbH, Kottingbrunn, Austria) and mini-injection molding (Thermo Scientific HAAKE Minijet, Waltham, MA, USA), respectively. During the microinjection molding, the injection speed is set at 300 mm/s or the corresponding volumetric flow rate of 5880 mm^3^/s. The dimension of the micropart is 18 × 3 × 0.3 mm^3^ (length × width × thickness). The mini-injection molding machine was controlled purely by pressure at 500 bar (the volumetric flow rate is 3.0 × 10^3^ mm^3^/s). The dimension of the macropart are 78 × 10 × 4 mm^3^ (length × width × thickness). The micropart and macropart were termed as the M-part and the C-part, respectively. The designations and formulations of the iPP/PET blends are listed in Figure 1 and Table 1. 

### 2.3. Characterization

#### 2.3.1. Differential Scanning Calorimetry (DSC)

The crystalline properties were tested by differential scanning calorimetry (DSC) named TA Q20 (TA Instruments, New Castle, DE, USA). About 5~8 mg samples were heated to 200 °C with the heating rate of 10 °C/min, and then held for 5 min to erase its thermal history. Finally, the samples were cooled to 40 °C, at a cooling rate of 10 °C/min. The relative crystallinity of β-form iPP, φ_β_, can be obtained by Equation (1) [23]:(1)$$\varphi_{\beta }=\frac{X_{\beta }}{X\alpha +X\beta }\times 100\%$$
where, X_α_ and X_β_ are the crystallinity for α- and β-form, respectively. Both values can be calculated through Equation (2):(2)$$X_{i}=\frac{\Delta H_{i}}{\Delta H_{i}^{\theta }}\times 100\%$$
where, X_i_ is the crystallinity of either the α- or β-form, $\Delta H_{i}$ is the specific fusion heat of the respective crystalline phase, $\Delta H_{i}^{\theta }$ is the standard fusion heat of both the α- and β-phase crystals of iPP, being 178 and 170 J/g, respectively [24,25].

#### 2.3.2. Scanning Electron Microscope (SEM)

To observe the crystalline structures of samples, the permanganic etching treatment was used to etch the sample, and then, the samples were tested through scanning electron microscopy (SEM) (JSM 840, JEOL, Tokyo, Japan).

#### 2.3.3. Polarized Light Microscope (PLM)

Injection molded parts were sectioned using a microtome for optical morphology comparation using a polarized light microscope (PLM) (Olympus BX51, Tokyo, Japan).

#### 2.3.4. Dynamic Rheology Measurements

The rotational rheometer (ARES, TA Instruments, New Castle, DE, USA) was used to observe the dynamic rheology properties. The blends were compressed into a disk with 25 mm diameter at the melt temperature of 270 °C. During the experiment, the gap and temperature were set at 1.5 mm and 270 °C, respectively. The shear rate was changed from 0.1 rads^−1^ to 450 rad s^−1^.

#### 2.3.5. Synchrotron Two-Dimensional Wide-Angle X-ray Diffraction (2D-WAXD) and Small-Angle X-ray Scattering (2D-SAXS)

The distribution of molecular orientation was conducted using the beamline BL16B1 at Shanghai Synchrotron Radiation Facility (SSRF, Shanghai, China). The orientation of iPP crystals is quantitatively calculated via the following orientation parameter ($f_{H}$) equation [26]:(3)$$f_{H}=\frac{\left(3\cos^{2}\varphi -1\right)}{2}$$
where, $\cos^{2}\varphi$ is an orientation factor defined as:(4)cos2φ=∫0π2I(θ)sinθcos2θdθ∫0π2I(θ)sinθdθ
where, θ is the azimuthal angle. Additionally, relative crystallinity of the β-form, K_β_, is calculated using Turner–Jones’ equation [27]:(5)$$K_{\beta }=\frac{I_{\beta 1}}{I_{\beta 1}+(I_{\alpha 1}+I_{\alpha 2}+I_{\alpha 3})}\times 100\%$$
where, I_β1_ is the diffraction of the β(300) plane and I_α1_, I_α2_, and I_α3_ are the diffraction of the α(110), α(040) and α(130) planes, respectively.

#### 2.3.6. Tensile Test

The mechanical properties of the samples were conducted vis a testing machine (Instron 4302, Instron Corporation, Turin, Italy) with a crosshead speed of 50 mm/min. Five specimens were tested for each measurement.

## 3. Results and Discussion

### 3.1. Dynamic Rheology Analyses

Before probing the changes of phase morphological structures, it is necessary to carry out dynamic rheology analyses of blends with varied PET concentration, as shown in Figure 2. It can be easily seen that, increasing the PET concentration can gradually decrease the complex viscosity in the low shear rate area. In the whole shear rate range, the complex viscosity of all of the specimens (except PET) was also decreased with the increasing the shear rate, exhibiting non-Newtonian behavior. 

In addition, Figure 2 also shows that the viscosity ratio of iPP/PET is less than 1, indicating that the PET phases maybe form the fibrillar morphology [28,29].

### 3.2. Morphological Observation

The changes of microstructure in iPP/PET blends and subsequent injection molded samples are presented in Figure 3 and Figure 4. Figure 3a–c shows that spherical PET domains are discretely distributed within iPP, which is a typical structure of immiscible polymer blends. Under the influence of intensive shearing conditions in μIM, the dispersed PET phase can be deformed and elongated into PET microfibrils along the FD in both the skin and core layers of subsequent M-parts, especially in terms of the PPET10 M-parts, as displayed in Figure 3f,f’. However, the microstructure in the CPPET10 C-parts is quite different. As can be seen from Figure 4, CPPET10 C-parts do not show a distinct change in the morphology of the dispersed PET phase in the core layer, while phase deformation slightly occurs in the skin layer along the FD, even under a much lower mold temperature (i.e., 80 °C). Such phenomenon arises from the differences of generated internal shear force fields and the temperature gradient of polymer melts during the mold cavity filling process. During the injection stage, the IM owns the lower shear rate (~10^2^ s^−1^) [30], resulting in the minor deformation of the PET phase. As the injection stage finishes, to reduce interfacial energy, the deformed PET particles are prior to return to their initial states (spherical states). Therefore, the elongated PET domains would have adequate relax time and transform into initial states after the injection stage, resulting from the lower shear stress and cooling rate in IM. In this scenario, the dispersed PET phase exhibits less deformation and elongation. However, this is not the case under the intensive shearing conditions in μIM. Compared with macroparts, microparts own the thinner thickness (~0.3 mm) and short filling time (~0.6 s) [6], resulting in the higher cooling rate (the cooling rate for M-parts is about 70 times quicker than that of C-parts). During the time in which the blends are injected into the mold cavity, the elongated PET domains can be frozen instantly, and the polymer chains cannot have enough time to recover to their original state through relaxation or reorientation. The mechanism of the above-mentioned phase evolution for both M-part and C-part is schematically displayed in Figure 5.

Interestingly, more highly oriented and well-defined PET microfibrils can be discerned in the core layer of the M-part, as shown in Figure 3d–f,d’–f’. The reason could be explained by the existing rheology, as well as the breakup and coalescence of dispersed phase [31] in μIM. According to literature [32], locations where there is a close proximity to the solidified layer or mold surface show much higher shear rates, whereas the positions at the center of the mold cavity channel exhibit the lowest shear rates. 

This suggests that the mold cavity wall owns the higher shear stress than that at the center. The higher shear stress exceeding the interfacial tension between iPP and PET phases can induce the breakup of long PET microfibrils into smaller entities, thereby leading to a poorer distribution of microfibrils in the skin layer of M-parts.

### 3.3. Crystalline Structure

The distribution of crystalline structure across the thickness direction in both μIM and IM samples was determined using PLM, as displayed in Figure 6 and Figure 7. As depicted in Figure 5, IM is characteristic of lower cooling and shear rates, prohibiting the relaxation of oriented structures. Hence, the ratio of the orientation layer (skin and shear layers) occupies a smaller portion across the thickness direction of the C-part. For example, the orientation layer of PPET0 M-parts is approximately four times thicker than that of IM counterparts, under the mold temperature of 120 °C. Moreover, a special cylindrulite structure was unexpectedly found in the core layer of the M-parts with 5% PET, as displayed in Figure 6b’. The oriented crystals are seemingly a shish-kebab structure, which are highlighted by yellow rectangles. According to the comparison between Figure 6a,b,a’,b’, typical crystalline structure cannot be easily discerned when they were prepared at a mold temperature of 80 °C. Furthermore, Figure 7 shows that the morphological distribution of crystalline structure in IM C-parts becomes more uniform, and the shish-kebab-like oriented structure cannot be formed when compared with the M-parts.

The most fascinating result from this work is that the shish-kebab-like β-crystals were found in the core region of the iPP/PET M-parts (see Figure 8). Therefore, the evolution of crystalline microstructure between the core and shear regions was examined across the thickness direction of samples prepared at T_mold_ = 120 °C. Two interesting findings can be discerned. Firstly, sparse cylindrites can be discerned along the surface of PET microfibrils in the core layer of iPP/PET M-parts, which can further be verified by the results of PLM. Secondly, the formation of PET fibrils greatly reduces the size of β-cylindrites, thereby leading to forming the shish-kebab-like β-cylindrites in the shear layer. The concerted effects of the presence of PET fibrils and the high shear rates on forming such crystalline structure are regarded as the driving factor. The self-seeding nucleation which caused the flow field favors the crystallization of the iPP chains by the existence of a high proportion of stretched chains. Furthermore, the specific mold temperature (120 °C) may promote the transition of α- to β-crystals [33,34]. Thus, the iPP chains near the surface of well-oriented PET microfibrils tend to fold into lamellar crystallites, resulting in forming brush-shaped transcrystalline structure, as displayed in Figure 8b,c. It thus can be concluded that the formation of transcrystalline structure stems from the synergetic effect of PET microfibrils, and the stretching of iPP chains.

### 3.4. Orientation Analysis

Figure 9 and Figure 10 show the 2D-WAXD reflection patterns of specimens. The distribution of diffraction rings at various locations in 2D-WAXD patterns represents different crystal planes. By carefully comparing with the M-part, it can be found that there is a visible difference in the diffraction rings of samples. Whether adding the PET phase or altering the mold temperature, the diffraction rings of the IM samples show more long-arc-like patterns in the skin layer, and even form circle-arc-like diffractions in the core layer, indicative of a lower level of molecular orientation.

Furthermore, the representative SAXS patterns which are inserted in Figure 9 and Figure 10 demonstrate two sharp, triangular streaks in the equatorial direction, and two bulb-shape lobes in the meridional direction, implying a reflection of shish-kebab superstructure that prevails in the M-parts and the skin layer of the C-parts [35,36]. In the core region of these C-parts, the diffuse signal is a full ring, suggesting a random distribution of crystal lamellae, and an absence of shish (i.e., row nuclei) in situ. According to Equations (3) and (4), the molecular orientation factor (*f**_WAXD_*) and lamellar orientation factor (*f**_SAXS_*) were obtained and listed in Table 2. The *f*_H_ of μ-parts exhibits an increasing trend with the content of PET, at the same mold temperature. During the cooling stage, the fast cooling rate caused by microinjection molding can rapidly freeze the chain, restraining the relaxation of this chain’s orientation. The synergistic effect of high shear stress and cooling rate can generate a high orientation degree (*f*_H_). Furthermore, this tendency becomes obvious with decreasing mold temperature (80 °C).

Interestingly, four (110) reflections around the meridian show a lamellar branching and twisting [37,38]. The similar lamellar branching was also observed in previous work, whereas the mechanism was indistinct [39]. From the integrated azimuthal scans of (110) in Figure 11, the curve of the (110) plane shows a bimodal maximum at 80° and 110°, and other maxima at 0°, which suggests the presence of a mixed bimodal orientation of α-crystals. These confirm that the branched lamella exists in the α-crystal, which is independent of the concentration of PET. Accordingly, the fraction of daughter lamellae (R) can be calculated using Equation (6) [40]:(6)$$R=\frac{A^{*}}{C+A^{*}}$$
where, *A** is taken as the area around an azimuthal angle of 90°, and C represents the area around 0° after subtracting the baseline area. The calculated results are 0.61 and 0.72 for PPET0 and PPET10 at the T_mold_ of 80 °C, respectively. The result indicates that α-crystals of the PPET10 M-parts possess a higher degree of branching. Moreover, with increasing T_mold_ (120 °C), the branching degree gradually increases from 0.61 to 0.75 for PPET0 and from 0.72 to 0.83 for PPET10, respectively.

### 3.5. Crystalline Structure Analysis

Figure 12 shows the DSC heating curves of the iPP/PET samples which were prepared by μIM and IM, respectively. The endothermic peak around 165 °C was observed, dur to the melting of α-crystals. When the mold temperature increased 120 °C, another small melting peak appeared at approximately 145 °C which corresponds to the melting of the β-crystals [41]. The intensity of β-crystals in the C-parts (except the case of CPPET0-S) becomes weaker and even vaguer when compared with that of the MIM counterparts. Relative to μIM, the polymer melt in IM has a much lower cooling rate due to its larger mold cavity. It was reported that the temperature gradient and higher shear stress can kinetically favor the transition of α to β crystals [32,34,42,43]. The shearing force field under the conditions of μIM is much stronger than that in IM, which can well explain the DSC results that the shear layer of both the M-parts and C-parts has a higher degree of β-crystals. In fact, the promoting effect was also verified by the results of 1D-WAXD profiles, as given in Figure 13 and Table 3. The content of β-crystals increases when the loading concentration of PET is no more than 5%, whereas it decreases when the content of PET exceeds 5%. This might be ascribed to the PET microfibrils having a heterogeneous nucleation ability and preserving the shear-induced orientation of the α-nuclei. Therefore, the amount of β cylindrulites decreases, and the crystal size becomes sparse. Table 3 shows that enhancing the T_mold_ and the content of PET can promote the formation of the β-crystal. The crystallization time of the M-parts increases significantly at the higher T_mold_. This extension of crystallization time results in an overgrowth of iPP chain segments to construct β-crystals. Therefore, higher mold temperature (i.e., 120 °C) is advantageous to generate β-form crystals.

### 3.6. Mechanical Properties

The tensile properties between iPP/PET M-parts and C-parts are illustrated in Figure 14. Figure 14 shows that the M-part demonstrates superior mechanical performance when compared with those of IM counterparts. As for C-parts, increasing mold temperature and the PET concentration from 0% to 5% can enhance the tensile strength from ~35 MPa to 40 MPa. In addition, applying extreme shear stress and increasing the mold temperature can further increase the tensile strength to ~46 MPa. The reason is that the abundantly dispersed PET phase can deform and emerge into microfibrils which act as a load transfer medium, and thereby contribute to enhancing the tensile property. With increasing the PET concentration from 5% to 10%, the tensile strength for microparts can be further improved from ~46 MPa to ~56 MPa. The possible explanations for this phenomenon are listed below. 

(1) The formation of an amount of fibers can act as load transfer agents, enhancing the tensile strength (seen Figure 3 and Figure 4). (2) The adhesion between the fiber and the matrix with the presence of bundle-like β-crystal surrounding PET fibers can be improved (Figure 8). (3) Introducing the PET microfibrils can enhance the orientation of molecular chains along the flow direction (seen Table 2). However, the elongation at break was reduced with an incremental loading fraction of PET. In this scenario, the immiscibility between PET and iPP may be the contributing factor. Additionally, mold temperature plays a negative role in ductility, especially in terms of μIM [40,41]. The reduction in elongation at break for samples molded at T_mold_ = 120 °C is likely related to the increased crystallinity, as described in Table 3. The above shows that the mechanical property can be greatly enhanced by the processing parameters, particularly in μIM, which is characteristic of high shearing and cooling effects. Therefore, it would be expected that microinjection-molded, immiscible polymer blends can be utilized in preparing high performance microdevices, which shows promising applications in the fields of electronics, automotive, biomedical and micro–electro–mechanical systems, among others. 

## 4. Conclusions

In this study, the properties between the microinjection-molded micropart (μIM M-part) and mini-injection molded micropart (IM C-part) were investigated in detail. Results showed that the μIM is characteristic of higher shearing and cooling effects when compared with the IM. The variation in thermomechanical history between both processing techniques leads to a totally different evolution of microstructure in subsequent samples. For example, the dispersed, spherical PET phases deform and emerge into well-defined microfibrils under the conditions of μIM, especially in the case of the PPET10 M-part. Moreover, the synergetic effect of the shear flow field and the presence of PET microfibrils aid the formation of bundle-like β-crystals. Samples demonstrated a higher orientation degree and crystalline structure when they were fabricated at lower mold temperature (i.e., 80 °C in this work). DSC and WAXD measurements reveal that increasing the content of PET leads to increasing content of the β-form and the degree of orientation in the M-part. 

In addition, the mechanical properties of the M-parts were superior to those of the IM C-parts, which is ascribed to the difference of the thermomechanical history between both processing methods. It is worth noting that this work promotes an alternative to achieve in-situ PET microfibrils, which can greatly improve the properties of μIM parts.

## Figures and Tables

**Figure 1 polymers-12-00219-f001:**
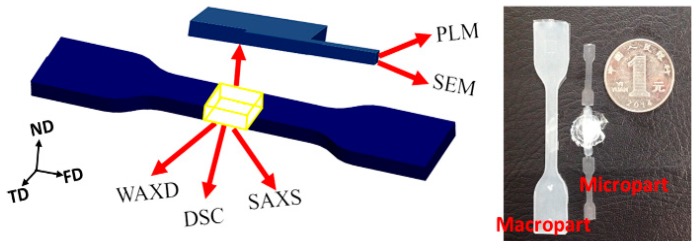
Schematic of samples for measurements. FD: flow direction, TD: transverse direction, ND: normal direction. SAXS: Small-Angle X-ray Scattering; DSC: differential scanning calorimetry; WAXD: Wide-Angle X-ray Diffraction; SEM: scanning electron microscopy; PLM: Polarized Light Microscope.

**Figure 2 polymers-12-00219-f002:**
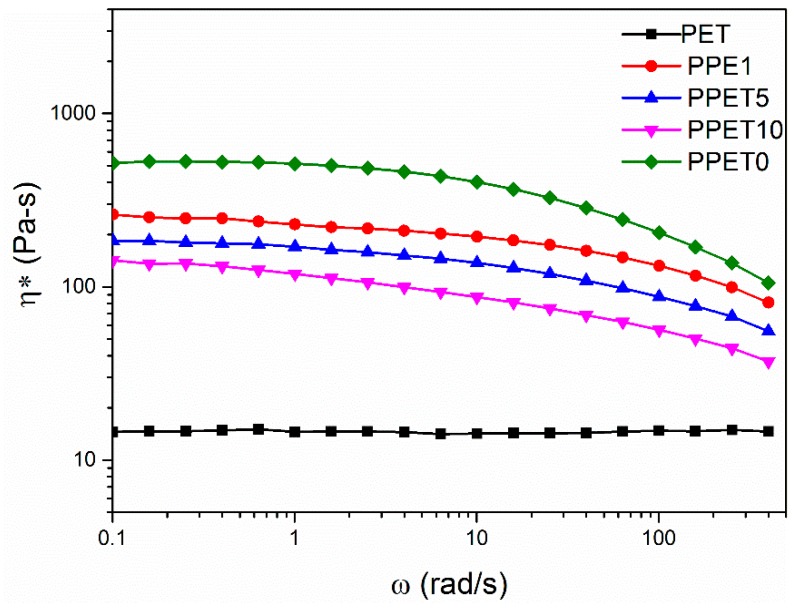
The changes of complex viscosity.

**Figure 3 polymers-12-00219-f003:**
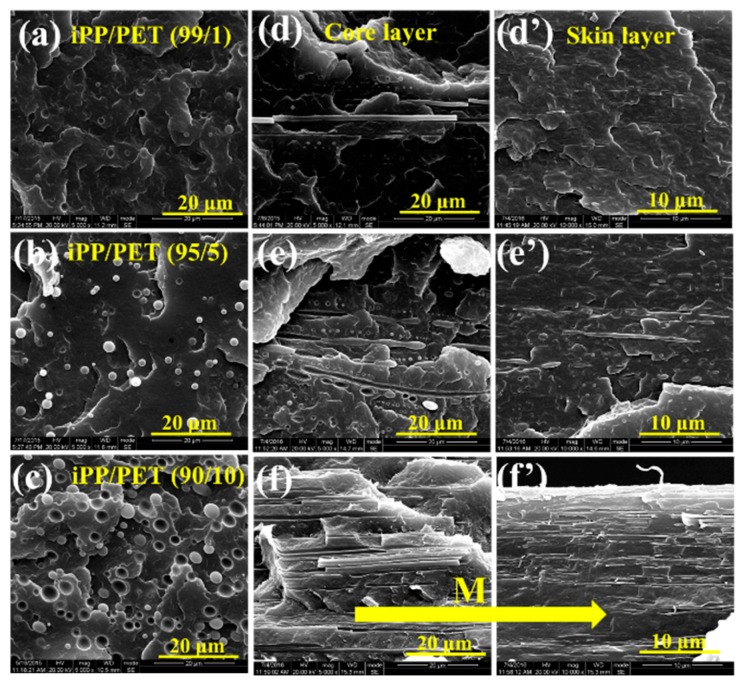
Scanning electron microscopy (SEM) images of iPP/PET blends: melt-blended specimens (**a**–**c**) and M-parts with (**d**–**f’**). The mold temperature is 120 °C. M represents the flow direction.

**Figure 4 polymers-12-00219-f004:**
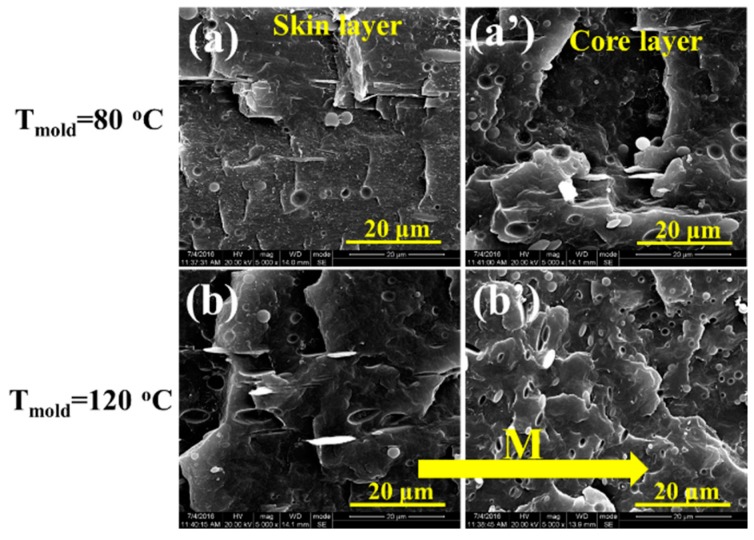
SEM images of iPP/PET (90/10) C-parts. M represents the flow direction.

**Figure 5 polymers-12-00219-f005:**
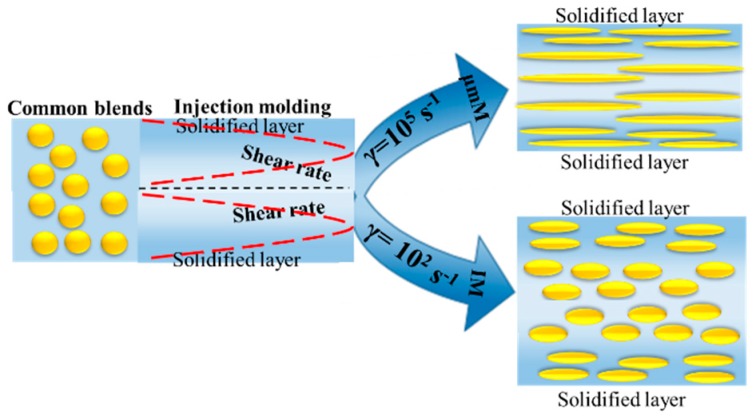
The mechanism of morphology evolution for iPP/PET blends in μIM and IM.

**Figure 6 polymers-12-00219-f006:**
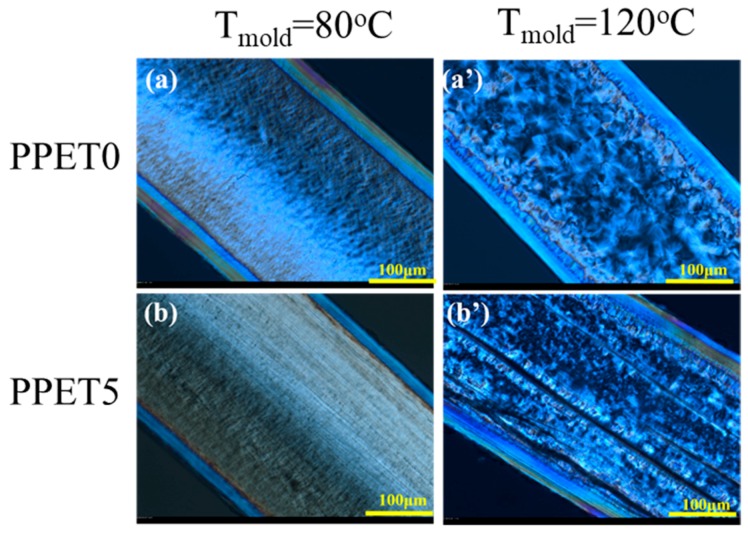
Polarized Light Microscope (PLM) images of crystalline structure in M-parts.

**Figure 7 polymers-12-00219-f007:**
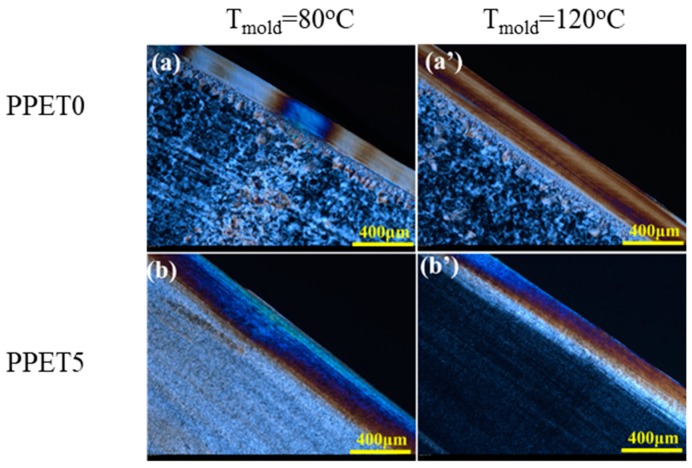
PLM images of crystalline structure in C-parts.

**Figure 8 polymers-12-00219-f008:**
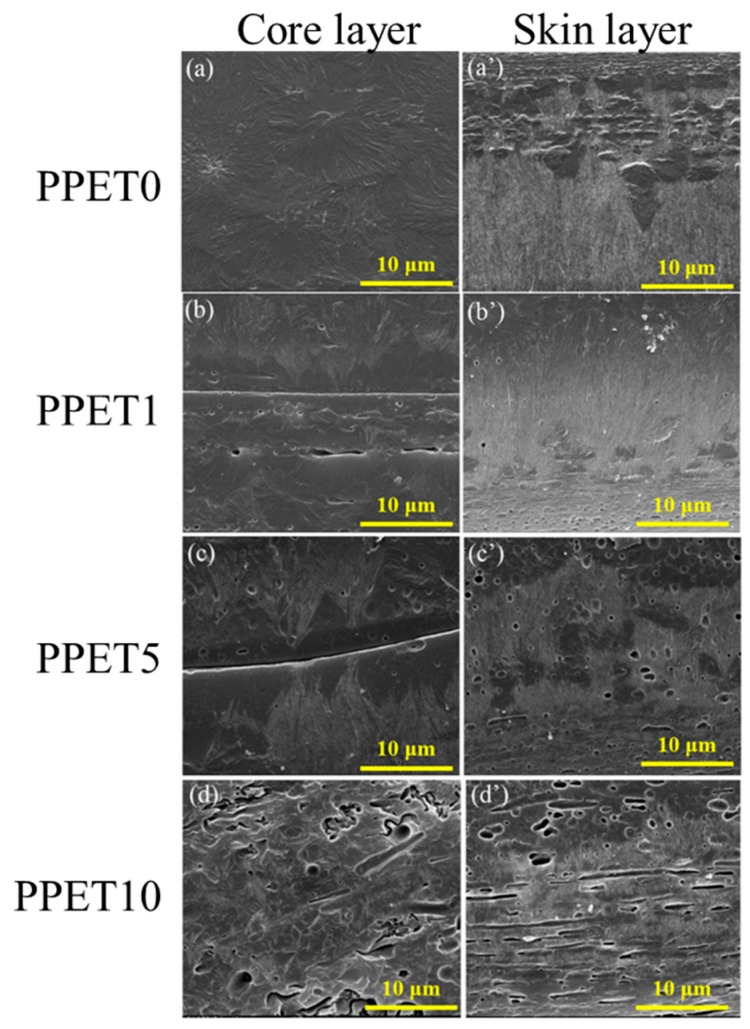
SEM images of M-parts. The mold temperature is 120 °C.

**Figure 9 polymers-12-00219-f009:**
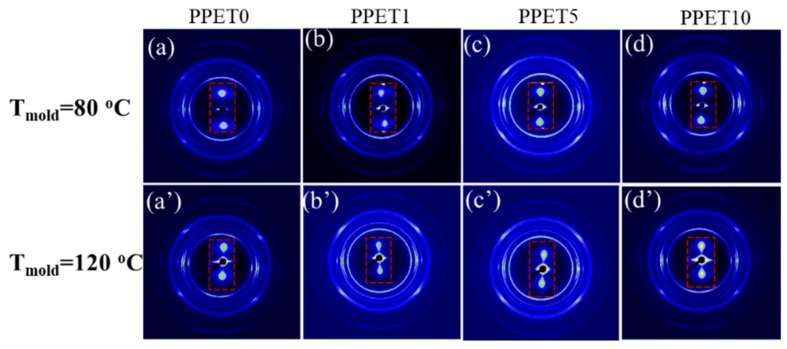
The two dimensional wide-angle X-ray diffraction (2D-WAXD) reflection patterns of the M-parts.

**Figure 10 polymers-12-00219-f010:**
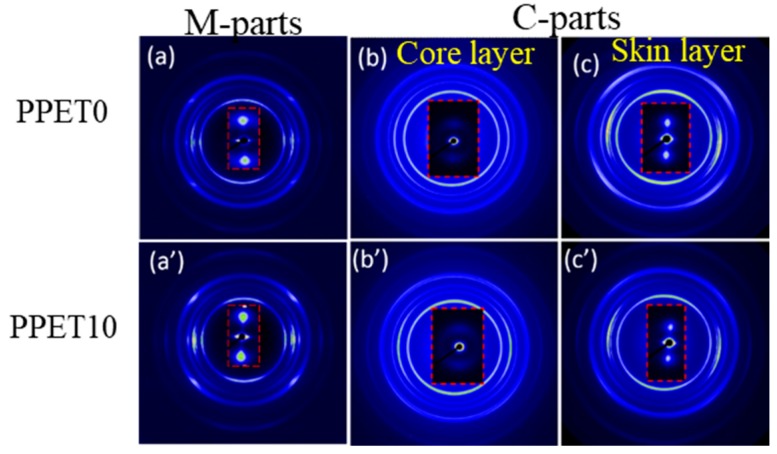
The 2D-WAXD reflection patterns samples at T_mold_ = 80 °C. The flow direction is the vertical direction.

**Figure 11 polymers-12-00219-f011:**
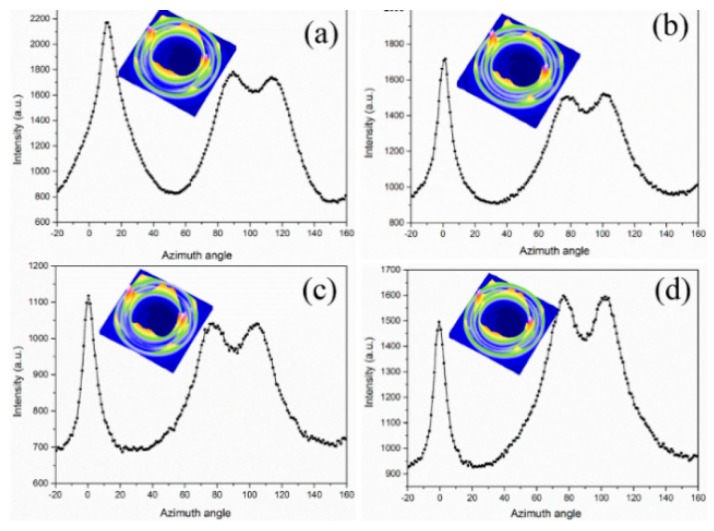
The azimuthal spreads of the (110) plane in different M-parts: (**a**,**c**): PPET0; (**b**,**d**): PPET10; (**a**,**b**): T_mold_ = 80 °C. (**c**,**d**): T_mold_ = 120 °C.

**Figure 12 polymers-12-00219-f012:**
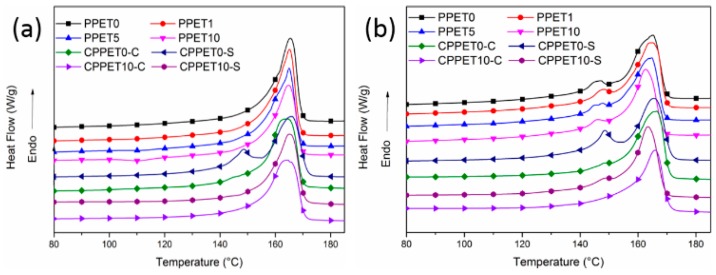
DSC melting curves of the samples: (**a**) T_mold_ = 80 °C; (**b**) T_mold_ = 120 °C; C represents the core layer; S represents the skin layer.

**Figure 13 polymers-12-00219-f013:**
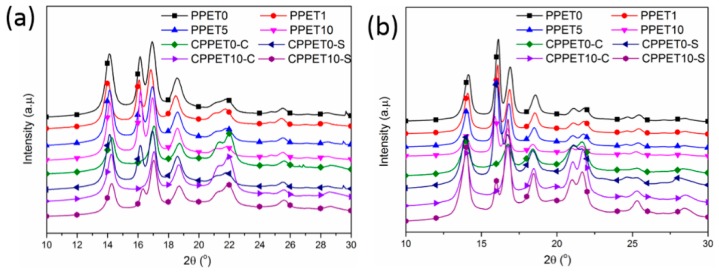
1D-WAXD curves of samples: (**a**) T_mold_ = 80 °C; (**b**) T_mold_ = 120 °C. C represents the core layer; S represents the skin layer.

**Figure 14 polymers-12-00219-f014:**
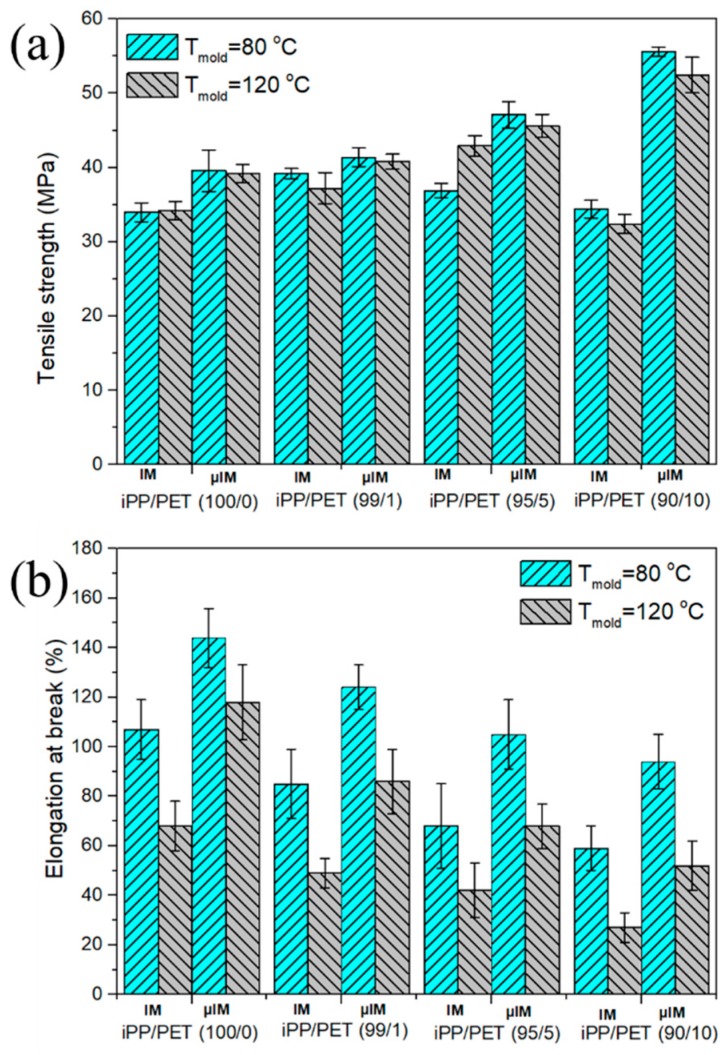
The tensile strength (**a**) and elongation at break (**b**) between M-parts and C-parts.

**Table 1 polymers-12-00219-t001:** Material designation and composition of isotactic polypropylene/poly(ethylene terephthalate) (iPP/PET) and the blends.

Designation	Composition
PPET0	iPP
PPET1	99 wt % iPP + 1 wt % PET
PPET5	95 wt % iPP + 5 wt % PET
PPET10	90 wt % iPP + 10 wt % PET

**Table 2 polymers-12-00219-t002:** Orientation degree fitted from 2D-WAXD and SAXS.

Samples	T_m_ = 80 °C		T_m_ = 120 °C	
	f_waxd_	f_saxs_	f_waxd_	f_saxs_
PPET0	0.912	0.856	0.874	0.826
PPET1	0.925	0.872	0.893	0.842
PPET5	0.956	0.937	0.926	0.897
PPET10	0.941	0.915	0.911	0.872

**Table 3 polymers-12-00219-t003:** Relative content of the β-crystal.

Samples	T_m_ = 120 °C		T_m_ = 80 °C	
	DSC	1D-WAXD	DSC	1D-WAXD
	K_β_	φ_β_	K_β_	φ_β_
PPET0	8.1	41.7	---	24.9
PPET1	8.4	42.8	---	28.1
PPET5	9.3	43.9	---	34.3
PPET10	7.9	39.6	---	32.7
CPPET0-C	4.2	--	---	---
CPPET0-S	21.3	46.9	---	17.5
CPPET10-C	---	---	---	---
CPPET10-S	--	13.6	---	7.8

Note: C represents the core layer; S represents the skin layer.

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
