# Peer review of "Crystallization and Microstructure Evolution of Microinjection Molded Isotactic Polypropylene with the Assistance of Poly(Ethylene Terephthalate)"

_polymers, 2020, doi:10.3390/polym12010219_

Round 1

Reviewer 1 Report

I have checked the manuscript and the authors' answers and I have decided to accept the paper as it is.

Reviewer 2 Report

In the latest version the text has been supplemented with several important aspects of the research. The text has also been supplemented with appropriate commentary and literature, so I believe that it can be published in its current form. 

Reviewer 3 Report

The authors have issued all the concerns, so I think it is suitable to publish this paper in Polymers.

This manuscript is a resubmission of an earlier submission. The following is a list of the peer review reports and author responses from that submission.

Round 1

Reviewer 1 Report

The work carried out by Zhao et al. investigated the influence of mold temperature on the microstructure, crystallization and tensile properties of iPP/PET blends.

Main issue:

It is surprising that authors did not cite the most relevant article entitled " Effects of Process Temperatures on the Flow-Induced Crystallization of Isotactic Polypropylene/Poly(ethylene terephthalate) Blends in Microinjection Molding, Ind. Eng. Chem. Res. 2017, 56, 9467−9477 " which is contributed by the same group of authors. In view of the above publication in which similar work has already been attempted at examining effect of mold temperature which results in the overall properties that are similar to that of the iPP/PET blends examined in the present study, nothing appears to be exciting.

Other comments

[1] The authors didn’t demonstrate enough novelty in this work. Therefore, the introduction should be rearranged and improved to show clearly the novelty of the present work.

[2] The authors stated in the abstract that" This study provided a new approach to fabricating  microdevices with enhanced mechanical properties". Thus, the tensile findings shown in Fig.12 should be discussed more and compared with previous investigations in literature. The lack of discussion makes difficult to determine the scientific contribution of the present work.

[3] I suggest to use same magnification in Figures 4 and 5. 

[4]   Please make all the captions of figures more informative.

[5] Where is the Equations (6 and 7)?

[6] The manuscript has been checked and found to contain excessive text from other publications amounting to more than 22%. It is understandable that it is difficult to completely avoid writing sentences similar to those already written, therefore, Similarity Index of about 10% can be tolerated.

In view of the aforementioned reasons, the article in its present form is not suitable for publication in Polymers.

Reviewer 2 Report

The structure/properties evaluation during the micro-injection molding process,  which is the subject of this paper is one of the more dynamically developed processing technique. Addressing this topic is therefore highly and can affect the further development of this method of production. The concept of using a PP / PET mixture to achieve desired nucleation effects is not new, however, in the context of micro injection molding, the level of novelty is significant. 

In my opinion the presented text is suitable for publication, however, some corrections can be performed.

1.Authors could present the appearance of the samples and/or mold cavity, this would give a better view of the MIM process idea and scale

2.It would be advisable to supplement the research with a rheological measurements, like rheometer tests (rotational or capillary) or even simple MFI measurements  

Reviewer 3 Report

The article deals with a rather interesting topic in the world of micro-manufacturing, namely the microstructure evolution on a polymeric component molded with the micro injection technique. The article is interesting and useful to enrich the micro molding background although it has some aspects that need strong improvement.

The abstract does not present how the research was designed but it presents directly the obtained results (since line 16) thus the reader has difficulties to focus the topic of the paper. I not agree with authors with the chosen notation for Micro Injection Molding (MIM) and Conventional Injection Molding (CIM) because, in many contexts, MIM stands for Metal Injection Molding and CIM for Ceramic Injection Molding. Thus, I suggest to use µIM for micro injection molding and IM for traditional injection molding. In the last two rows of the abstract the authors “promise” that “the study provided a new approach to fabricating microdevices with enhanced mechanical properties”. I think that it was not maintained because the authors presented many results goings in this direction, but they must summarize all the findings. To accomplish this, I suggest to add a paragraph about discussion after the section 3.5. Lines 32-34: I think that somethings are missing because the sentence is not clear and there is the lack of a dot. I suggest to the authors to add the design of the realised part but above all they must add the details about the used molding machines. In fact, the paper presents the differences obtained on samples as effect of the two processes (micro and conventional molding) but these effects are due to the machines and especially these are effects of the applied process parameters whose settings were not discussed. Furthermore, how the authors select the parameters that were studied (mold temperature and composition)? I think that also other process parameters (above all the injection velocity) strongly influence the obtained microstructures. Moreover, the molding condition and setting for both the processes must be reported. How was the shear rate (reported in figure 3) calculated? The molded part is the same in the two processes and also the material. The only difference between micro and mini machine is the injection speed but there is no mention about it. The paper deals about micro molding but the realised part in not really micro but it is a mini thin part. How do you justify this? In the experimental section there is no mention about the number of samples that were molded and if some replications were done to obtained more reliable results. Only in the lines 126-127, about tensile tests, the authors report that five specimens were tested. Could you clarify this aspect? Lines 140: the discussion about shear rate needs clarification as reported in previous consideration about injection velocity. Also the consideration about cooling reported in line 147 should be better explained. Fig. 1 should be enlarged.